# *Wolfram syndrome 1* regulates sleep in dopamine receptor neurons by modulating calcium homeostasis

Huanfeng Hao[1], Li Song[1], Luoying Zhang[1,2]*

1 Key Laboratory of Molecular Biophysics of Ministry of Education, College of Life Science and Technology, Huazhong University of Science and Technology, Wuhan, Hubei, China, 2 Hubei Province Key Laboratory of Oral and Maxillofacial Development and Regeneration, Wuhan, Hubei, China

These authors contributed equally to this work.
* zhangluoying@hust.edu.cn

**Data Availability Statement:** All relevant data are within the manuscript and its Supporting Information files.

**Funding:** This work was supported by grants from the Ministry of Science and Technology of China

## Abstract

Sleep disruptions are quite common in psychological disorders, but the underlying mechanism remains obscure. Wolfram syndrome 1 (WS1) is an autosomal recessive disease mainly characterized by diabetes insipidus/mellitus, neurodegeneration and psychological disorders. It is caused by loss-of function mutations of the *WOLFRAM SYNDROME 1* (*WFS1*) gene, which encodes an endoplasmic reticulum (ER)-resident transmembrane protein. Heterozygous mutation carriers do not develop WS1 but exhibit 26-fold higher risk of having psychological disorders. Since WS1 patients display sleep abnormalities, we aimed to explore the role of WFS1 in sleep regulation so as to help elucidate the cause of sleep disruptions in psychological disorders. We found in *Drosophila* that knocking down *wfs1* in all neurons and *wfs1* mutation lead to reduced sleep and dampened circadian rhythm. These phenotypes are mainly caused by lack of *wfs1* in dopamine 2-like receptor (Dop2R) neurons which act to promote wake. Consistently, the influence of *wfs1* on sleep is blocked or partially rescued by inhibiting or knocking down the rate-limiting enzyme of dopamine synthesis, suggesting that *wfs1* modulates sleep via dopaminergic signaling. Knocking down *wfs1* alters the excitability of Dop2R neurons, while genetic interactions reveal that lack of *wfs1* reduces sleep via perturbation of ER-mediated calcium homeostasis. Taken together, we propose a role for *wfs1* in modulating the activities of Dop2R neurons by impinging on intracellular calcium homeostasis, and this in turn influences sleep. These findings provide a potential mechanistic insight for pathogenesis of diseases associated with *WFS1* mutations.

## Author summary

Psychiatric disorders are often accompanied by sleep disruptions, but the causes of theses disruptions are largely unclear. Wolfram syndrome 1 (WS1) is a neurodegenerative disease co-morbid with psychiatric disorders. It is caused by homozygous mutation of the *WOLFRAM SYNDROME 1* (*WFS1*) gene, which encodes a transmembrane protein that

(STI 2030-Major Projects 2021ZD0203200-02), as well as Natural Science Foundation of China (31930021 and 32022035) to LZ. The funders had no role in study design, data collection and analysis, decision to publish, or preparation of the manuscript.

**Competing interests:** The authors have declared that no competing interests exist.

localizes to the endoplasmic membrane (ER). Heterozygous mutation carriers do not develop WS1 but display 26-fold higher risk for having psychiatric disorders. WS1 patients also experience sleep abnormalities, therefore here we investigate the role of WFS1 in sleep regulation in hope that this will help us better understand the sleep disruptions associated with psychiatric disorders. Using the fruit fly *Drosophila*, we found that *wfs1* deficiency in wake-promoting dopamine receptor neurons leads to reduced sleep and dampened circadian rhythm. Consistently, pharmacological treatment and genetic interaction analysis reveal that the sleep-modulating role of *wfs1* requires dopamine. Furthermore, we found that the sleep reduction caused by *wfs1* deficiency is likely due to alteration of ER-mediated calcium homeostasis, which in turn influences neural activity and neurotransmitter release. These findings highlight a role for dopamine and dopamine receptor neurons in WFS1-related sleep disturbances, which may serve as a common mechanism for sleep disruptions associated with psychiatric disorders.

## Introduction

Sleep disruptions are common in individuals with psychiatric disorders, and sleep disturbances are risk factors for future onset of depression [1]. However, the mechanism underlying sleep disruptions in psychiatric disorders are largely unclear. Wolfram Syndrome 1 (WS1) is an autosomal recessive neurodegenerative disease characterized by diabetes insipidus, diabetes mellitus, optic atrophy, deafness and psychiatric abnormalities such as severe depression, psychosis and aggression [2–4]. It is caused by homozygous (and compound heterozygous) mutation of the *WOLFRAM SYNDROME 1* (*WFS1*) gene, which encodes wolframin, an endoplasmic reticulum (ER) resident protein highly expressed in the heart, brain, and pancreas. On the other hand, heterozygous mutation of *WFS1* does not lead to WS1 but increase the risk of depression by 26 fold [5,6]. A study in mice further confirmed that *WFS1* mutation is causative for depression [7]. Consistent with the comorbidity of psychiatric conditions and sleep abnormalities, WS1 patients also experience increased sleep problems compared to individuals with type I diabetes and healthy controls [8]. It has been proposed that sleep symptoms can be used as a biomarker of the disease, especially during relatively early stages, but the mechanisms underlying these sleep disturbances are unclear. Considering that heterozygous *WFS1* mutation is present in up to 1% of the population and may be a significant cause of psychiatric disorder in the general population, we decided to investigate the role of wolframin in sleep regulation so as to probe the mechanism underlying sleep disruptions in psychiatric disorders [5,6,9].

Although the wolframin protein does not possess distinct functional domains, a number of *ex vivo* studies in cultured cells demonstrated a role for it in regulating cellular responses to ER stress and calcium homeostasis, as well as ER-mitochondria cross-talk [10–13]. Mice that lack *Wfs1* in pancreatic β cells develop glucose intolerance and insulin deficiency due to enhanced ER stress and apoptosis [14–16]. Knocking out *Wfs1* in layer 2/3 pyramidal neurons of the medial prefrontal cortex in mice results in increased depression-like behaviors in response to acute restraint stress. This is accompanied by hyperactivation of the hypothalamic-pituitary-adrenal axis and altered accumulation of growth and neurotrophic factors, possibly due to defective ER function [7]. A more recent study in *Drosophila* found that knocking down *wfs1* in the nervous system does not increase ER stress, but enhances the susceptibility to oxidative stress-, endotoxicity- and tauopathy-induced behavioral deficits and neurodegeneration [17]. Overall, the physiological function of wolframin *in vivo*, especially in the brain, remains elusive for the most part.

Here we identified a role for wolframin in regulating sleep and circadian rhythm in flies. *Wfs1* deficiency in the dopamine 2-like receptor (Dop2R) neurons leads to reduced sleep, while inhibiting dopamine synthesis blocks the effect of *wfs1* on sleep, implying that *wfs1* influences sleep via dopaminergic signaling. We further found that these Dop2R neurons function to promote wakefulness. Depletion of *wfs1* alters neural activity, which leads to increased wakefulness and reduced sleep. Consistent with this, we found that knocking down the ER calcium channel *Ryanodine receptor* (*RyR*) or *1,4,5-trisphosphate receptor* (*Itpr*) rescues while knocking down the sarco(endoplasmic)reticulum ATPase *SERCA* synergistically enhances the short-sleep phenotype caused by *wfs1* deficiency, indicating that wolframin regulates sleep by modulating calcium homeostasis [18]. Taken together, our findings provide a potential mechanism for the sleep disruptions associated with *WFS1* mutation, and deepen our understanding of the functional role of wolframin in the brain.

## Results

### *Wfs1* deficiency leads to reduced sleep and dampened locomotor rhythm

A previous study reported that wolframin functions in both neurons and glial cells to regulate climbing ability, life span and neurodegeneration [17]. Therefore, to investigate the consequence of *wfs1* deficiency on sleep and circadian rhythm, we knocked down *wfs1* in all cells, neurons and glial cells using *tubulin*(*tub*)-GAL4, *elav*-GAL4 and *repo*-GAL4, respectively [19,20]. Depletion of *wfs1* in all cells and neurons result in substantially reduced sleep duration and power of locomotor rhythm which indicates dampened circadian rhythm, while the period of the rhythm is not altered (Fig 1A–1D and Table 1). These phenotypes are consistent between the two independent RNAi lines used, both of which lead to significant reduction of *wfs1* mRNA level (S1A Fig). On the other hand, knocking down *wfs1* in glial cells did not significantly alter sleep or locomotor rhythm, strongly suggesting that wolframin acts in neurons to regulate sleep and locomotor rhythm (Fig 1E and 1F and Table 1). To ensure that the phenotypes we observed in *wfs1* RNAi flies are not merely due to over-expression of RNAi, we adopted a GFP RNAi as a control. Expressing GFP-RNAi does not significantly alter sleep or locomotor rhythm (S1B Fig and Table 1).

To further validate that the sleep and circadian disruptions are indeed due to *wfs1* deficiency, we assessed the phenotypic effects of *wfs1^MI14041^* mutation which is expected to produce a C-terminal truncated wolframin and may act as a partial loss of function mutation [17]. Homozygous *wfs1^MI14041^* mutants also show significantly reduced sleep and power, consistent with the RNAi results (Fig 1G and 1H and Table 1). Given that the sleep reduction appears to be more prominent in male mutants compared to females, we focused on males for the remainder of this study. To test whether neuronal wolframin expression is sufficient for maintaining sleep and locomotor rhythm, we expressed *wfs1* in the neurons of *wfs1^MI14041^* mutants and found that both the reduced sleep and dampened rhythm were rescued in these flies (Figs 1I and S1C and Table 1). Taken together, these series of results indicate that neuronal wolframin is both necessary and sufficient for sleep and locomotor rhythm.

Next, we systematically analyzed the effects of knocking down *wfs1* pan-neuronally on sleep structure. *wfs1* RNAi lead to decreased sleep both in the light and dark phases, while waking activity is not significantly altered, indicating that the short-sleep phenotype is not caused by hyper-activity (Fig 1J). The sleep reduction is a result of decreased sleep bout length but not bout number, which means that *wfs1* depletion perturbs sleep maintenance but not sleep initiation. Consistently, sleep latency is not altered in *wfs1* RNAi flies, further implicating that sleep initiation system is intact.

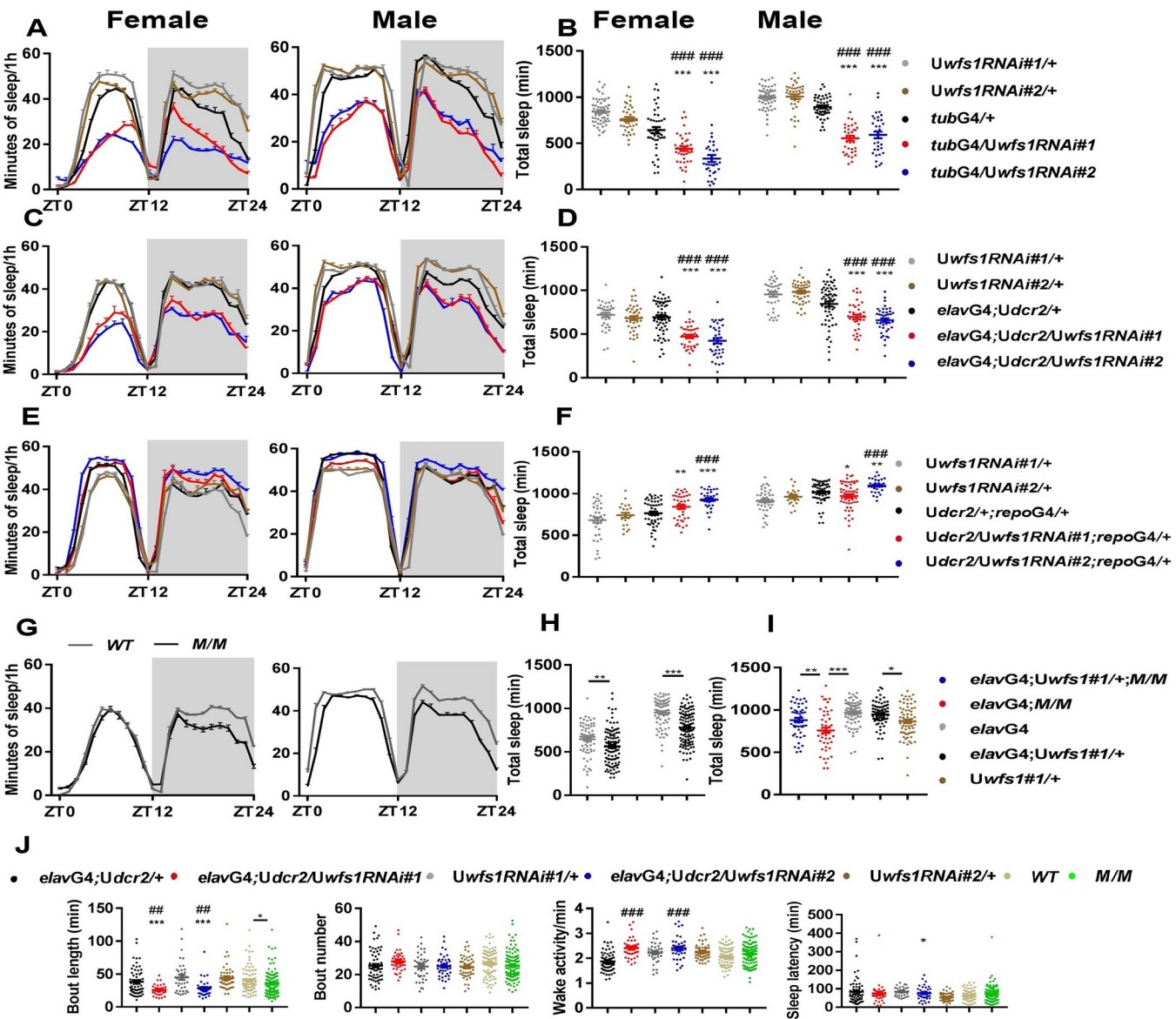

**Fig 1. *Wfs1* deficiency leads to reduced sleep and dampened locomotor rhythm.** The sleep of female and male flies are monitored under 12 h light: 12 h dark (LD) condition. (A, C, E and G) Sleep profile of *wfs1* RNAi, mutant and control flies. Gray shade indicates the dark period. (B, D, F and H) Daily sleep duration of *wfs1* RNAi, mutant and control flies. (I) Daily sleep duration of *wfs1* mutant, pan-neuronal rescue and control flies. (J) Sleep bout length, bout number, wake activity and sleep latency of *wfs1* RNAi, mutant and control flies. For comparison between RNAi or over-expression flies vs. UAS/GAL4 controls, one-way ANOVA was used: compared to GAL4 control, ##$P < 0.01$, ###$P < 0.001$; compared to UAS control, *$P < 0.05$, **$P < 0.01$, ***$P < 0.001$. For comparison between mutant vs. control, Mann-Whitney test or Student's t-test was used: *$P < 0.05$, **$P < 0.01$, ***$P < 0.001$. For comparison between rescue vs. control, one-way ANOVA was used: **$P < 0.01$. n = 20–80. Error bars represent standard error of the mean (SEM). G4, GAL4; U, UAS; *M*, *wfs1^{MI14041}*; ZT, Zeitgeber Time (ZT0 is the time of lights on).

## *Wfs1* deficiency during development leads to reduced sleep and dampened locomotor rhythm in adults

Neurodegeneration-related symptoms start to occur within the first decade of WS1 patients, with an average age of development ranging from 11–16 years [2–4]. Therefore, we investigated whether the sleep and circadian phenotypes observed in flies that lack *wfs1* are caused by deficiency during developmental and/or adult stage. We used a temperature sensitive *tubulin*

**Table 1. *Wfs1* deficiency leads to dampened locomotor rhythm.**

| Genotype | Period±SEM (hour) | Power±SEM | Rhythmicity | N |
|---|---|---|---|---|
| *Uwfs1RNAi#1/+* | 23.71±0.03 | 63.20±3.82 | 95% | 75 |
| *Uwfs1RNAi#2/+* | 23.69±0.03 | 58.70±4.04 | 95% | 65 |
| *tub*G4/+ | 24.12±0.04 | 74.3±5.99 | 88% | 57 |
| *tub*G4/Uwfs1RNAi#1 | 24.25±0.13 | 8.00±2.41*** ### | 30% | 27 |
| *tub*G4/Uwfs1RNAi#2 | 24.28±0.17 | 22.17±5.43** ### | 60% | 15 |
| *elav*G4 | 23.84±0.04 | 49.42±3.47 | 83% | 77 |
| *elav*G4;Uwfs1#1/+;MM | 23.71±0.05 | 59.13±5.69$ $ $ | 88% | 48 |
| *elav*G4;M/M | 23.53±0.11 | 18.00±4.51### | 43% | 42 |
| *elav*G4;Uwfs1#1/+ | 23.78±0.06 | 74.85±5.41## | 97% | 35 |
| Uwfs1/+ | 23.65±0.04 | 69.75±7.89 | 86% | 35 |
| *elav*G4;Uwfs1RNAi#1/+ | 23.87±0.06 | 24.35±3.46*** ### | 70% | 43 |
| *elav*G4;Uwfs1RNAi#2/+ | 23.96±0.11 | 10.24±2.53*** ### | 33% | 36 |
| *Udcr2/+;repo*G4/+ | 23.92±0.11 | 56.51±7.12 | 91% | 47 |
| *Udcr2/Uwfs1RNAi#1;repo*G4/+ | 23.69±0.07 | 72.62±7.45 | 100% | 24 |
| *Udcr2/Uwfs1RNAi#2;repo*G4/+ | 23.79±0.08 | 64.19±6.04 | 100% | 34 |
| *Udcr2/+;cry*G4-16/+ | 24.90±0.09 | 46.04±7.16 | 86% | 28 |
| *Udcr2/Uwfs1RNAi#1;cry*G4-16/+ | 24.46±0.08 | 65.14±5.37 | 100% | 26 |
| *Udcr2/Uwfs1RNAi#2;cry*G4-16/+ | 24.50±0.06 | 63.70±6.61 | 88% | 32 |
| *tim*G4/+ | 24.44±0.04 | 47.15±6.06 | 76% | 33 |
| *tim*G4/wfs1RNAi#1 | 24.33±0.06 | 66.63±7.51 | 84% | 31 |
| *tim*G4/wfs1RNAi#2 | 24.36±0.04 | 76.38±6.72 | 93% | 43 |
| **WT** | 23.73±0.04 | 50.80±4.85 | 82% | 62 |
| *M/M* | 23.62±0.04 | 33.31±4.38@@@ | 47% | 95 |
| *Dop2R*G4 | 23.70±0.04 | 62.19±4.74 | 83% | 84 |
| *Dop2R*G4;Uwfs1RNAi#1/+ | 23.78±0.05 | 29.69±3.66*** ### | 57% | 90 |
| *Dop2R*G4;Uwfs1RNAi#2/+ | 24.16±0.32 | 21.35±3.41*** ### | 43% | 82 |
| Gαo*G4/+ | 23.85±0.09 | 36.60±5.35 | 70% | 37 |
| Gαo*G4/Uwfs1RNAi#1 | 23.72±0.15 | 7.55±2.86 *** ### | 32% | 28 |
| Gαo*G4/Uwfs1RNAi#2 | 24.00±0 | 1.41±0.73 *** ### | 5% | 21 |
| *UGFPRNAi/+* | 23.61±0.04 | 75.71±10.04 | 95% | 23 |
| *tub*G4/UGFPRNAi | 24±0.03 | 60.97±7.37 | 82% | 34 |
| *elav*G4;Udcr2/+ | 23.78±0.07 | 76.02±8.63 | 100% | 20 |
| *elav*G4;Udcr2/UGFPRNAi | 23.82±0.07 | 77.96±9.19 | 91% | 24 |
| *Udcr2/UGFPRNAi;repo*G4/+ | 23.78±0.07 | 88.82±8.88## | 100% | 32 |

One-way ANOVA, #/*$P$ < 0.05, ##/**$P$ < 0.01, ###/***$P$ < 0.001. Mann-Whitney test, ###/$ $ $/@@@$P$ < 0.001. # compared with the GAL4 controls; * compared with the UAS controls; $ compared with GAL4 under mutant background; @ compared with WT. G4, GAL4; U, UAS; M, wfs1^{MI14041}.

(*tub*)GAL80$^{ts}$ to knock down *wfs1* specifically during the adult or developmental stage [21]. *tub*GAL80$^{ts}$ represses the transcriptional activities of GAL4 at permissive temperature (18˚C), thus GAL4-driven transcription can only occur under restrictive temperature (29˚C). When RNAi expression is restricted to adults, neither sleep duration nor locomotor rhythm is significantly altered (S2A Fig and S1 Table). On the other hand, when RNAi is expressed exclusively during the developmental stage, sleep is significantly reduced (S2B Fig). We noticed that when flies are raised under 29˚C, the adult locomotor rhythm is dampened even for some of the control lines (S1 Table). Therefore, we could not assess the influence of knocking down *wfs1* during development on locomotor rhythm.

Previous study has shown that *wfs1* deficiency results in significant behavioral deficits and neurodegeneration in flies by 30 days of age [17]. Therefore, we also examined the effects of *wfs1* deficiency on sleep and locomotor rhythm in aged flies. We found that only one of the RNAi lines when expressed leads to significantly decreased sleep, and the extent of this reduction appears to be smaller than that of younger flies (S3A–S3D Fig). Aged *wfs1* mutants display comparable decrease in sleep compared to that of younger flies (S3E and S3F Fig). Significant reduction of the power of locomotor rhythm can be observed in aged *wfs1* RNAi and mutant flies, but the power values are relatively low even for controls which is due to aging-induced dampening of circadian rhythm (S2 Table). Overall, the sleep and locomotor rhythm phenotypes associated with *wfs1* deficiency do not appear to be more severe as animals age.

Taken together, these series of findings indicate that *wfs1* is required during development for flies to maintain normal sleep and locomotor rhythm when they are adults.

### *Wfs1* deficiency reduces sleep independent of circadian clock and light/ dark condition

Because of the dampened locomotor rhythm in flies lacking *wfs1*, we tested whether circadian disruption contributed to the decreased sleep in these flies. We subjected *wfs1* RNAi and mutant flies to constant light (LL) condition which rapidly evokes circadian behavioral arrhythmicity and breakdown of the molecular clock [22,23]. We were still able to observe substantial reduction in sleep under LL for both RNAi and mutant flies, indicating that wolframin regulates sleep in a manner that is independent of the circadian clock (Fig 2A and 2B). Since it has been shown that the detrimental effects of LL on the rhythm lasts for a few days into constant darkness (DD), we transferred the flies to DD after LL and again observed decreased sleep in *wfs1* RNAi and mutant flies (Fig 2C and 2D) [24]. Taken together, these results demonstrate that the influence of wolframin on sleep does not require the clock or light/darkness. To further validate this, we measured the mRNA levels of core clock genes *period* (*per*), *timeless* (*tim*) and *clock* (*clk*), as well as the circadian photoreceptor *cryptochrome* (*cry*) [25]. There is no significant difference between *wfs1* RNAi and control flies (Fig 2E).

In addition, we investigated whether wolframin alters sleep homeostasis by assessing recovery sleep after 24h sleep deprivation and observed no significant change between RNAi flies and controls (S4 Fig).

### Wolframin maintains sleep and locomotor rhythm via dopaminergic signaling

To probe the mechanism by which *wfs1* deficiency influences sleep, we adopted a pharmacological approach and treated flies with drugs that target neurotransmitter systems known to regulate sleep, as well as the glutamate release inhibitor riluzole which has been shown to suppress the lifespan phenotype of *wfs1* RNAi flies (S5 Fig) [17,26–28]. We found that tyrosine hydroxylase inhibitor α-methyl-para-tyrosine (AMPT) could eliminate the effects of *wfs1* deficiency on sleep both in RNAi and mutant flies (Fig 3A–3C and 3E–3G). Further analysis reveals that AMPT increases sleep in a dose-dependent manner, while this increase is more prominent in *wfs1* deficient flies (Fig 3D and 3H). These results suggest that the influence of wolframin on sleep requires dopaminergic signaling.

To further validate this, we knocked down the gene encoding tyrosine hydroxylase (*ple*) using a previously published RNAi line [29]. *ple* deficiency partially rescues the short-sleep phenotype of *wfs1* mutants (Fig 3I). On the other hand, knocking down *ple* fully rescues the reduced power of locomotor rhythm in *wfs1* mutants (S3 Table). We also examined the effects

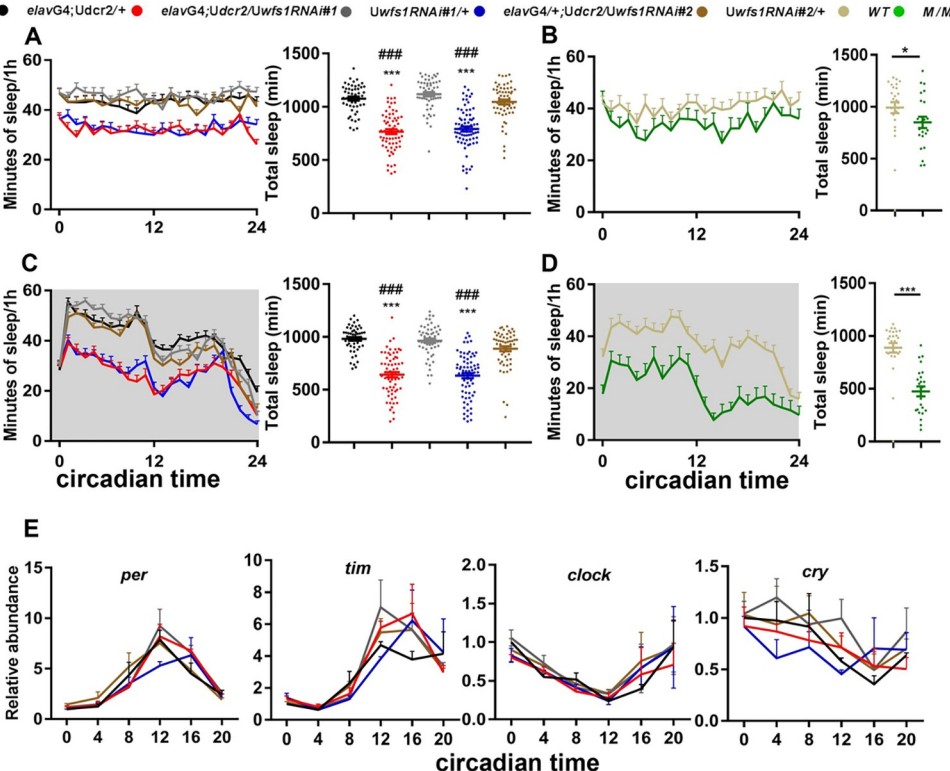

**Fig 2. *Wfs1* deficiency reduces sleep independent of circadian clock and light/dark condition.** (A and B) The sleep profile (left panel) and daily sleep duration (right panel) of *wfs1* RNAi (A) and mutant flies (B) monitored on the 4th day of constant light, along with relevant controls. (C and D) The sleep profile (left panel) and daily sleep duration (right panel) of *wfs1* RNAi (C) and mutant flies (D) monitored on the first day of constant darkness (DD), along with relevant controls. Gray shade indicates the dark period. For comparison between RNAi flies vs. UAS/GAL4 controls, one-way ANOVA was used: compared to GAL4 control, ###$P < 0.001$; compared to UAS control, ***$P < 0.001$. For comparison between mutant vs. control, Mann-Whitney test was used: *$P < 0.05$, ***$P < 0.001$. n = 23–73. (E) Plots of relative mRNA abundance vs circadian time (CT0, circadian time 0 is the time of subjective lights on) for clock genes determined by qRT–PCR in whole-head extracts. n = 3. For each time series, the value of G4 control at CT0 was set to 1. Error bars represent SEM. G4, GAL4; U, UAS; *M*, *wfs1*[MI14041].

of knocking down *wfs1* on the mRNA level of dopamine-related genes and dopamine concentration, but observed no significant difference (S6 Fig).

All in all, these results support the notion that wolframin function to regulate sleep and locomotor rhythm via dopaminergic signaling.

## Wolframin acts in Dop2R neurons to maintain sleep via Dop2R signaling

Next, we tested whether wolframin functions in dopaminergic and/or dopamine receptor neurons to regulate sleep. We knocked down *wfs1* in dopaminergic neurons using *dopamine transporter* (*DAT*)-GAL4 and *pale* (*ple*)-GAL4, as well as in various dopamine receptor neurons [30]. We found that lack of *wfs1* in Dop2R neurons reduces sleep (Fig 4A and 4B). Because Dop2R has been reported to be a receptor coupled to G$_o$α protein, we knocked down *wfs1* in G$_o$α-expressing cells using GAL4 line NP3200 and found this also resulted in reduced sleep (Fig 4A and 4C) [31,32]. We further expressed *wfs1* in Dop2R neurons of *wfs1*[MI14041] mutants and found this rescued the decreased sleep, indicating that wolframin expression in Dop2R neurons is both necessary and sufficient for sleep regulation (Fig 4D).

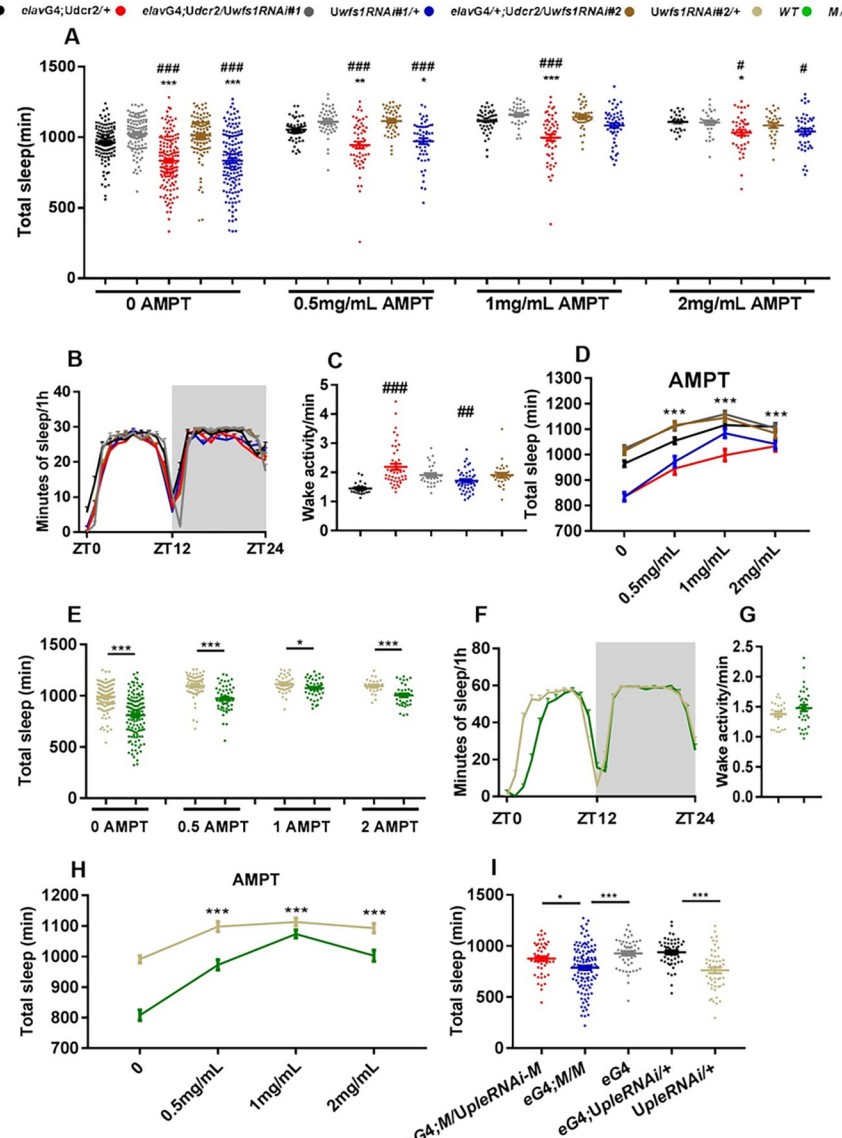

**Fig 3. Wolframin maintains sleep via dopaminergic signaling.** (A) Daily sleep duration of *wfs1* RNAi and control flies fed with different concentrations of AMPT. (B) The sleep profile and (C) wake activity of *wfs1* RNAi and control flies fed with 2mg/mL AMPT. (D) The dosage effect of AMPT on daily sleep duration of *wfs1* RNAi and control flies. (E) Daily sleep duration of *wfs1* mutant and control flies fed with different concentrations of AMPT. (F) The sleep profile and (G) wake activity of *wfs1* mutant and control flies fed with 2mg/mL AMPT. (H) The dosage effect of AMPT on daily sleep duration of *wfs1* mutant and control flies. (I) Daily sleep duration of *wfs1* mutant flies with pan-neuronal knock down of *ple*. Gray shade indicates the dark period. For comparison between RNAi flies vs. UAS/GAL4 controls, one-way ANOVA was used: compared to GAL4 control, #$P < 0.05$, ##$P < 0.01$, ###$P < 0.001$; compared to UAS control, *$P < 0.05$, **$P < 0.01$, ***$P < 0.001$. For comparison between mutant vs. control, Mann-Whitney test or Student's t-test was used: *$P < 0.05$, ***$P < 0.001$. n = 25–155. For comparison between mutant expressing RNAi vs. control, one-way ANOVA was used: *$P < 0.05$. Error bars represent SEM. *e*G4, *elav*GAL4; G4, GAL4; U, UAS; M, *wfs1*^MI14041^; ZT, Zeitgeber Time.

Consistent with the effects of wolframin on sleep, knocking down *wfs1* in Dop2R neurons or $G_{o}\alpha+$ cells reduces the power of locomotor rhythm, while knocking down *wfs1* in clock cells using *tim*GAL4 and *cry*GAL4-16 do not affect the rhythm (Table 1) [33,34]. Taken together with the result that pan-neuronal knock down of *wfs1* does not alter clock gene

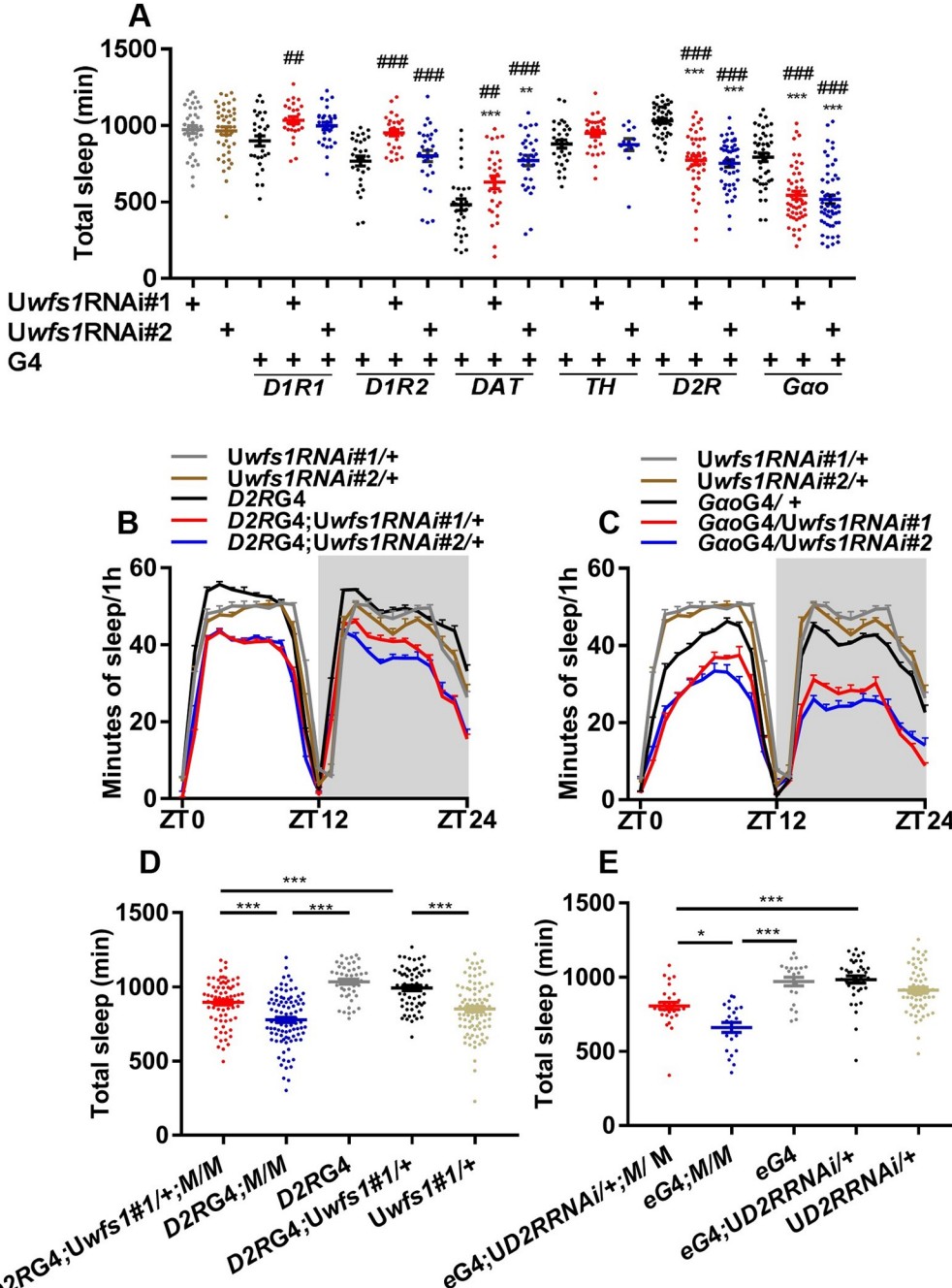

**Fig 4. Wolframin acts in Dop2R neurons to maintain sleep via Dop2R signaling.** (A) Daily sleep duration of flies with *wfs1* knocked down in dopaminergic neurons, dopamine receptor neurons and Goα+ cells. (B and C) The sleep profile of flies with *wfs1* knocked down in Dop2R neurons (B) or Goα+ cells (C). (D) Daily sleep duration of *wfs1* mutant, Dop2R neuron rescue and control flies. (E) Daily sleep duration of *wfs1* mutant flies with pan-neuronal knock down of *Dop2R*. Gray shade indicates the dark period. For comparison between RNAi or over-expression flies vs. UAS/ GAL4 controls, one-way ANOVA was used: compared to GAL4 control, ##$P < 0.01$, ###$P < 0.001$; compared to UAS control, *$P < 0.05$, **$P < 0.01$, ***$P < 0.001$. For comparison between mutant vs. control, Mann-Whitney test or Student's t-test was used: *$P < 0.05$, ***$P < 0.001$. For comparison between *wfs1* rescue vs. controls, one-way ANOVA was used: ***$P < 0.001$. For comparison between mutant expressing RNAi vs. controls, one-way ANOVA was used: *$P < 0.05$, ***$P < 0.001$. n = 13–94. Error bars represent SEM. *e*GAL4, *elav*GAL4; *D2R*, *Dop2R*; G4, GAL4; U, UAS; *M*, *wfs1*^MI14041^; ZT, Zeitgeber Time.

expression (Fig 2E), we believe that wolframin acts downstream of the clock in the circadian output pathway to regulate locomotor rhythm.

Lastly, we tested whether Dop2R signaling mediates the effects of wolframin on sleep and locomotor rhythm. We found that knocking down *Dop2R* using a previously published RNAi partially rescues the short-sleep phenotype of *wfs1* mutant, but does not significantly alter locomotor rhythm (Fig 4E and S3 Table) [32]. Taken together, these results suggest that wolframin acts in Dop2R neurons and maintains sleep by down-regulating Dop2R signaling.

## Dop2R and $G_o\alpha$+ cells act to promote wakefulness

Since the role of Dop2R neurons or $G_o\alpha$-expressing cells in sleep regulation has not been previously reported, we expressed the temperature-gated depolarizing cation channel TrpA1 in these cells to assess the effects of activating them [35]. Activating Dop2R neurons by a higher temperature lead to dramatically reduced sleep, indicating that these cells function to promote wakefulness (Fig 5A and 5B). On the other hand, waking activity is rather decreased when these neurons are activated, which means the enhanced wakefulness is not merely due to hyper activity (Fig 5C). Consistently, activating the $G_o\alpha$+ cells with TrpA1 also resulted in much decreased sleep while waking activity is rather decreased (Fig 5A–5C). In addition, we employed an alternative method to activate these cells by using the bacterial depolarization-activated sodium channel NachBac [36]. Flies expressing NachBac in Dop2R or $G_o\alpha$+ cells did not survive to adulthood. To resolve this issue, we used *tub*GAL80^ts in combination with *Dop2R*GAL4 or $G_o\alpha$GAL4 to express NachBac specifically during the adult stage. NachBac-expressing flies show significantly reduced sleep while waking activity is decreased (Fig 5D–5I).

These results indicate that Dop2R and $G_o\alpha$+ cells act to promote wakefulness, and thus knocking down *wfs1* in these cells may enhance their activities, leading to increased wakefulness and decreased sleep.

## *Wfs1* deficiency decreases sleep by altering the activity of Dop2R neurons

Multiple studies in cultured cells reported roles for wolframin in regulating cellular responses to ER stress and calcium homeostasis, while on the other hand, it has been shown that *wfs1* depletion in the fly nervous system does not enhance ER stress [10–13,17]. Based on these work, we speculate that lack of *wfs1* results in altered calcium centration in Dop2R/$G_o\alpha$+ cells which in turn leads to altered activity and function. To test this, we used a nuclear factor of activated T cells (NFAT)-based CaLexA reporter [37]. In this system, GFP is expressed in response to sustained neural activity. Since both *Dop2R*GAL4 and $G_o\alpha$GAL4 are expressed in the mushroom body (MB), an important brain structure for sleep and wake regulation, we focused on the effects of *wfs1* deficiency in the MB (S7 Fig) [38–40]. Strikingly, we observed dramatically elevated GFP signals in the MB of *wfs1* RNAi flies, implicating increased activity in these neurons (Fig 6A and 6B). To further validate this, we used synapto-pHluorin (spH) which is a fluorescent indicator of neurotransmitter release [41]. spH is localized to synaptic vesicles and fluorescence is increased in a pH-dependent manner when vesicles fuse with the presynaptic membrane. spH fluorescence is significantly increased in MB Dop2R neurons of *wfs1* RNAi flies, indicative of enhanced synaptic release likely due to elevated excitability (Fig 6C).

We also used the calcium sensor GCaMP6m to monitor intracellular calcium levels [42]. We found that pan-neuronal knock-down of *wfs1* significantly reduces GCaMP signal in MB neurons during the earlier half of the day (S8A and S8B Fig). This effect appears to be selective, as GCaMP signal is not altered in the antennal lobe as a control (S8C Fig). Consistent with behavioral data, AMPT treatment blocks the effect of *wfs1* deficiency on GCaMP signal

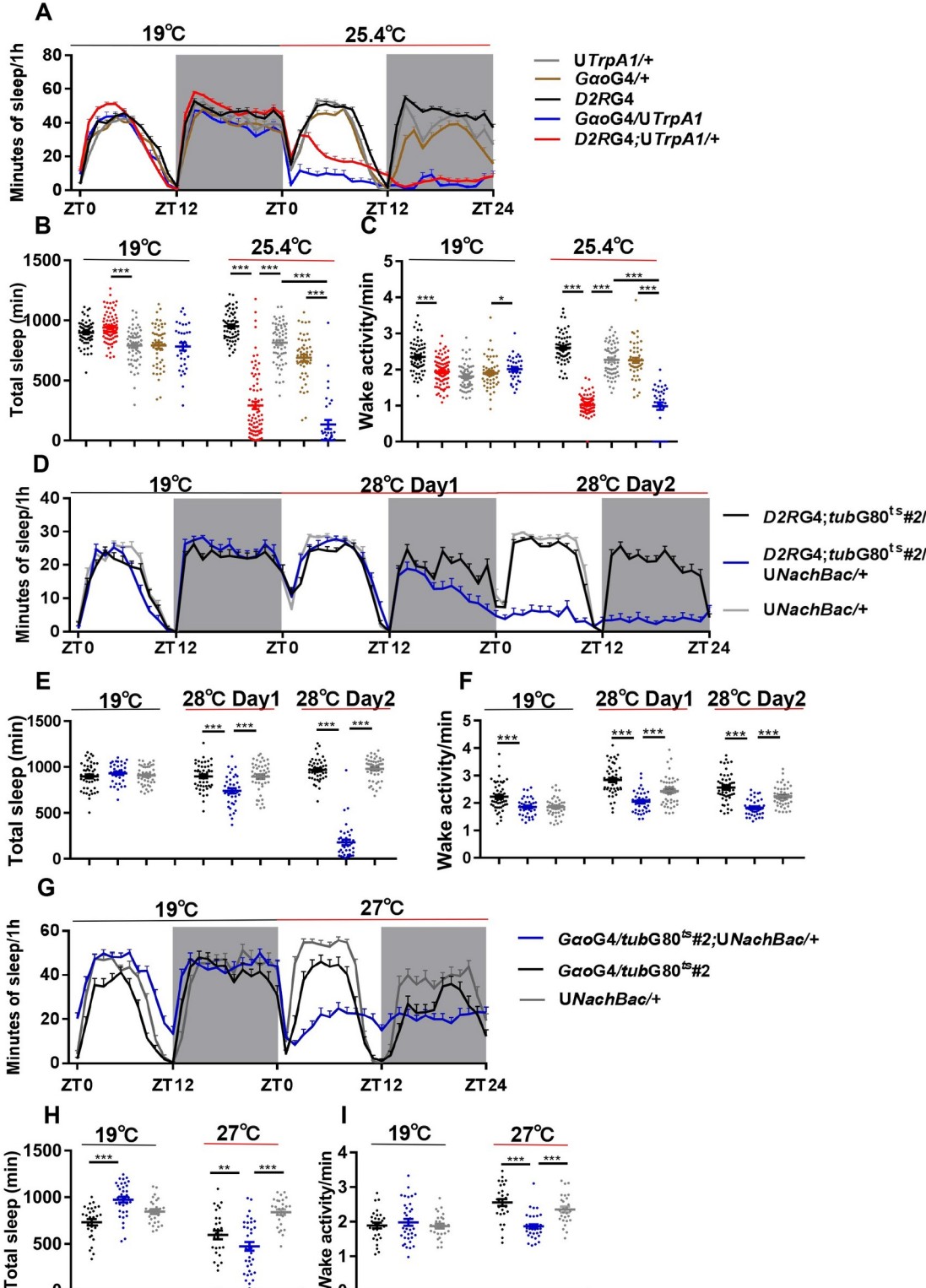

**Fig 5. Dop2R and $G_o\alpha+$ cells act to promote wakefulness.** (A-C) The sleep profile (A), daily sleep duration (B), and wake activity (C) of TrpA1 expressing flies and controls under baseline temperature (19°C) and activation temperature (25.4°C). (D-F) The sleep profile (D), daily sleep duration (E), and wake activity (F) of flies expressing NachBac in Dop2R neurons and controls under baseline temperature (19°C) and activation temperature (28°C). (G-I) The sleep profile (G), daily sleep duration (H), and wake activity (I) of flies expressing NachBac in $G_o\alpha+$ cells and controls under baseline temperature (19°C) and activation temperature

(27˚C). Gray shade indicates the dark period. One-way ANOVA: *$P < 0.05$, **$P < 0.01$, ***$P < 0.001$. n = 23–80. Error bars represent SEM. *D2R*G4, *Dop2R*GAL4; G4, GAL4; U, UAS; ZT, Zeitgeber Time.

(S8D Fig). In addition, we observed reduced GCaMP signal in MB Dop2R neurons of *wfs1* RNAi flies (S8E Fig). Moreover, we examined the effect of knocking down *wfs1* on dopamine-induced changes of GCaMP signal. We found that dopamine elicits a significantly smaller increase of GCaMP signal in Dop2R neurons of *wfs1* RNAi flies (S8F Fig and S1 and S2 Movies).

To determine whether the altered neural activity of Dop2R+ MB neurons is the cause of reduced sleep in *wfs1* deficient flies, we first adopted a MB247GAL80 which inhibits GAL4 activity in MB (S9A Fig) [43]. This attenuates the short-sleep phenotype of *wfs1* RNAi flies, which indicates that lack of *wfs1* in MB contributes to the decreased sleep in these flies (S9B Fig). We next tested whether silencing Dop2R neurons could rescue the short-sleep phenotype of *wfs1* mutants by expressing a temperature-sensitive mutant dynamin (*shi^{ts1}*) which blocks synaptic vesicle recycling at restrictive temperature (29˚C) [44]. While *wfs1* mutants expressing *shi^{ts1}* display reduced sleep under permissive temperature (19˚C), this sleep phenotype is rescued under restrictive temperature (Fig 6D). Consistently, over-expressing *wfs1* can rescue the decreased sleep caused by activating Dop2R neurons (Fig 6E). In addition, we tested for genetic interaction between activation of Dop2R neurons and *wfs1* heterozygous mutation but did not observe significant synergistic effect (S10 Fig).

Taken together, these results indicate that the sleep loss associated with *wfs1* deficiency is caused by altered excitability of Dop2R neuron.

## *Wfs1* interacts with genes that control ER calcium store to regulate sleep

Given the effects of *wfs1* deficiency on intracellular calcium level which should largely reflect changes of cytoplasmic calcium level, we next tested whether this altered cytoplasmic calcium homeostasis is the cause of sleep reduction [45]. Pharmacological experiments suggested that wolframin may modulate cytoplasmic calcium level via Ryanodine receptor (RyR), a gated calcium channel that controls the release of calcium from ER [46]. Therefore, we tested for genetic interaction between *wfs1* and *RyR* in sleep regulation. We used two independent RNAi lines to knock down *RyR* in Dop2R neurons of *wfs1* mutant flies, and found that one line (RNAi#1) fully rescued the short sleep phenotype while the other line (RNAi#2) partially rescued this phenotype (Figs 7A and S11A). On the other hand, knocking down *RyR* alone does not result in altered sleep duration. We also conducted pan-neuronal knock-down in *wfs1* mutants using these two RNAi lines. While RNAi#1 does not produce progenies when expressed pan-neuronally, RNAi#2 leads to a substantial lengthening of sleep duration when expressed pan-neuronally (Fig 7B).

Previous study has also implicated that inositol 1,4,5-trisphosphate (IP3) receptor (IP3R), another channel that releases calcium from ER, mediates the influences of *WFS1* deficiency on ER calcium homeostasis [47,48]. Therefore, we tested for genetic interaction between *wfs1* and *Itpr* which encodes IP3R. Similar to *RyR*, knocking down *Itpr* pan-neuronally using a previously published RNAi line leads to a substantial lengthening of sleep duration in *wfs1* mutants (S12 Fig) [49].

Since the genetic interactions between *RyR/Itpr* and *wfs1* suggest that *RyR* and *Itpr* function in opposite direction of *wfs1* to regulate sleep, we also tested for genetic interaction between *wfs1* and *SERCA*, which pumps calcium into ER and has been previously shown to directly bind with wolframin [50]. In contrary to *RyR* and *Itpr*, we found that knocking down *SERCA* (with one of the two independent RNAi lines, RNAi #2) in Dop2R neurons enhances the

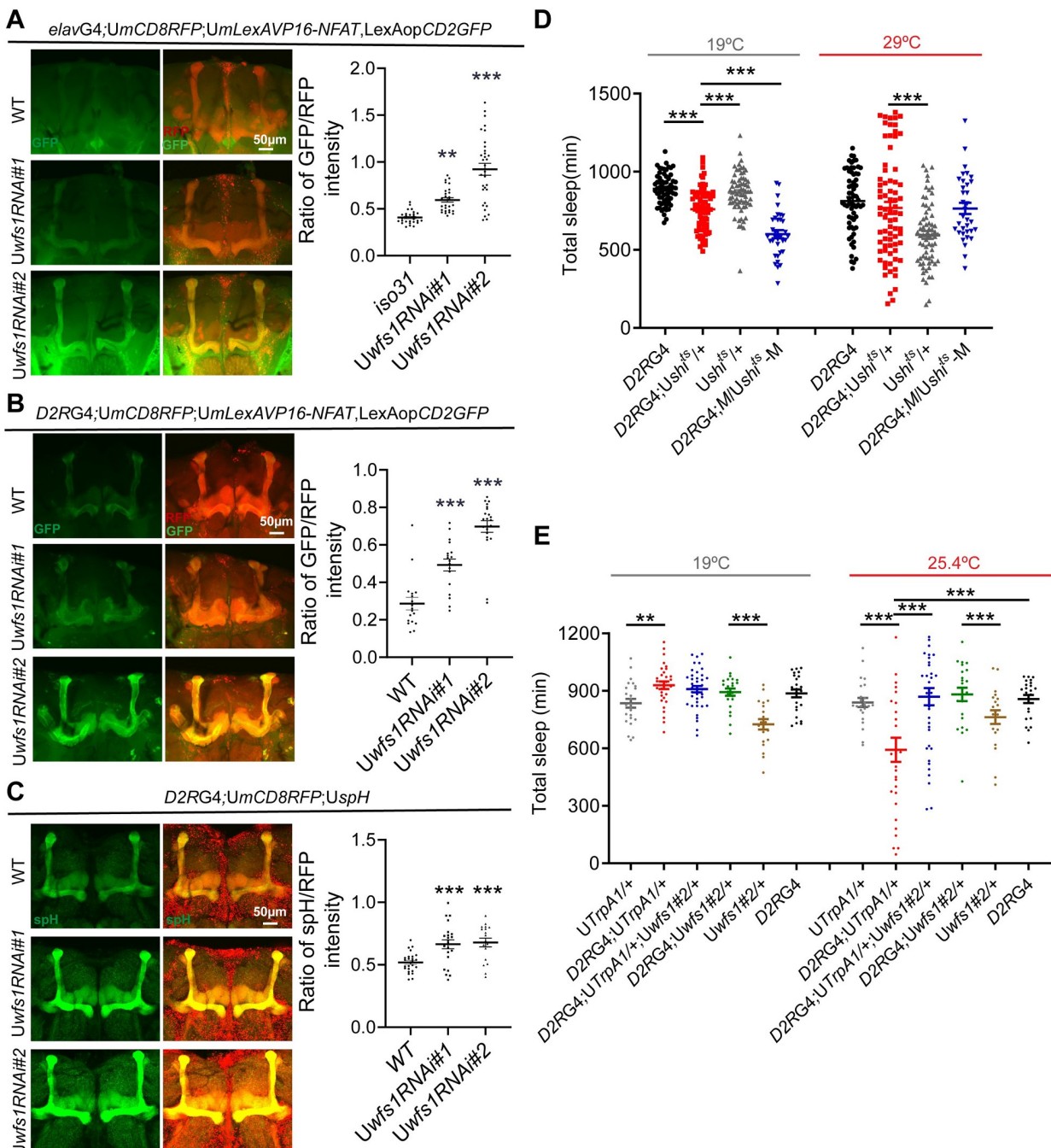

**Fig 6. *Wfs1* deficiency decreases sleep by altering the activity of Dop2R neurons.** (A) Left panel: representative images of adult fly brain with pan-neuronal expression of CaLexA and mCD8RFP. Right panel: quantification of GFP/RFP signal intensity in MB (two-way ANOVA, n = 28, 28, 32). (B) Left panel: representative images of adult fly brain expressing CaLexA and CD8RFP in Dop2R neurons. Right panel: quantification of GFP/RFP signal intensity in MB (two-way ANOVA, n = 18, 18, 22). (C) Left panel: representative images of adult fly brain with spH expression in Dop2R neurons. Right panel: quantification of spH/RFP signal intensity in MB (two-way ANOVA, n = 26, 22, 18). (D) Daily sleep duration of *shi*[ts1]-expressing *wfs1* mutant flies and controls under permissive temperature (19˚C) and restrictive temperature (29˚C). n = 34–75. (E) Daily sleep duration of flies over-expressing *wfs1* and *TrpA1*, as well as controls under baseline temperature (19˚C) and activation temperature (25.4˚C). n = 20–39. One-way ANOVA: *$P < 0.05$, **$P < 0.01$, ***$P < 0.001$. Error bars represent SEM. The scale bar represents 50 μm unless indicated otherwise. G4, GAL4; U, UAS; *D2R*G4, *Dop2R*GAL4.

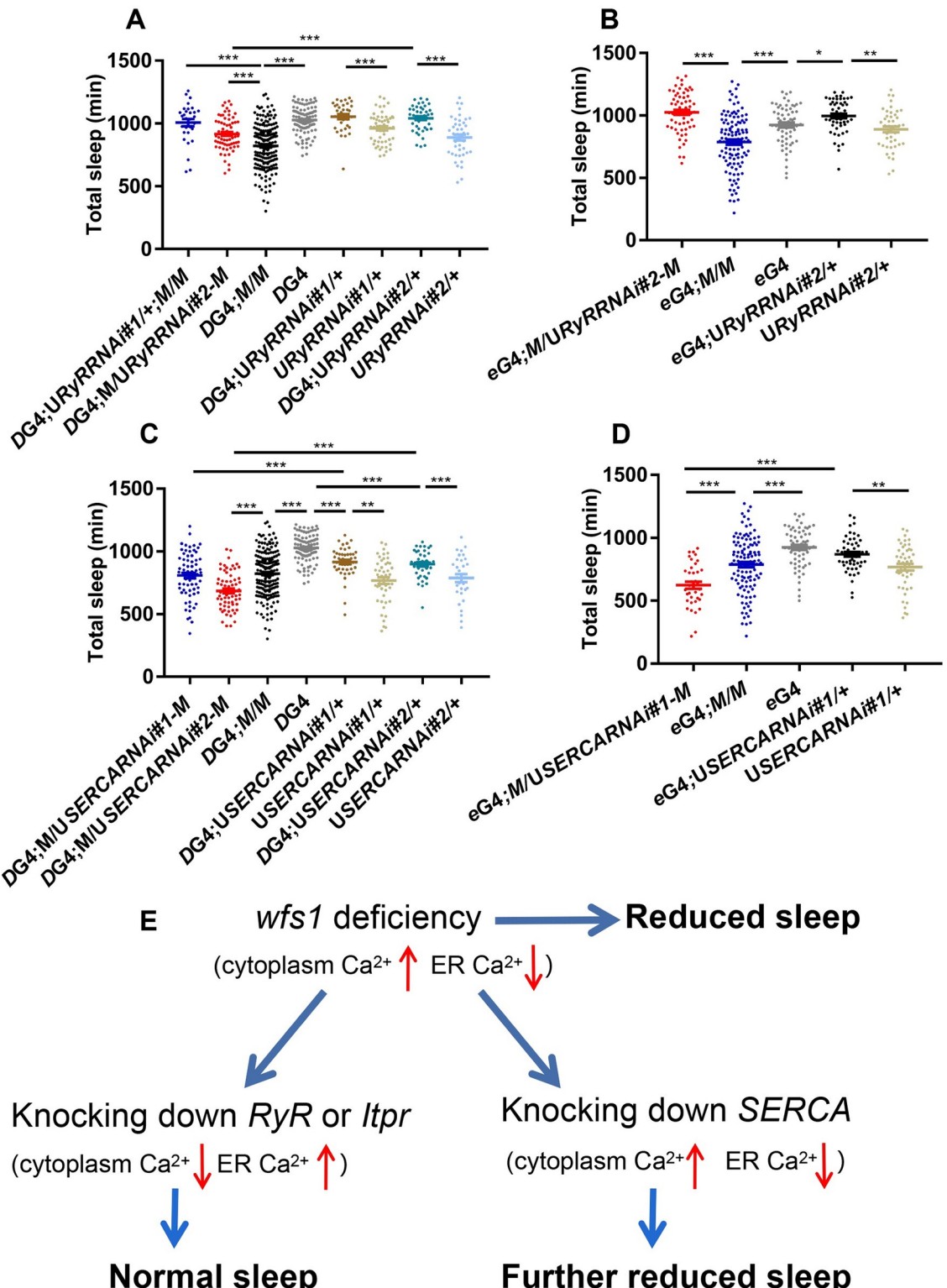

**Fig 7. *Wfs1* interacts with genes that control ER calcium store to regulate sleep.** (A and B) Daily sleep duration of *wfs1* mutant flies with *RyR* knocked down in Dop2R neurons (A) or all neurons (B). (C and D) Daily sleep duration of *wfs1* mutant flies with *SERCA* knocked down in Dop2R neurons (C) or all neurons (D). For comparison between RNAi flies vs. UAS/GAL4 controls, one-way ANOVA was used. For comparison between mutants vs. controls, Student's t-test was used. *$P < 0.05$, **$P < 0.01$, ***$P < 0.001$. For comparison between mutant expressing RNAi vs. control, one-way ANOVA was used: ***$P < 0.001$. n = 30–195. Error bars

represent SEM. *DG4*, *Dop2R*GAL4; *e*G4, *elav*GAL4; U, UAS; M, *wfs1^{MI14041}*. (E) Model illustrating the genetic interactions between *wfs1* and *RyR/Itpr/SERCA*.

short-sleep phenotype of *wfs1^{MI14041}*, while knocking down *SERCA* alone does not alter sleep duration (Figs 7C and S11B). For panneuronal knock-down we were not able to obtain flies expressing RNAi#2, while expression of RNAi#1 results in reduction of sleep on *wfs1* mutant but not wild-type background (Fig 7D).

Taken together, these results demonstrate that *RyR* and *Itpr* deficiency rescues or reverses the sleep reduction in *wfs1* mutants, while *SERCA* deficiency synergistically enhances the sleep reduction. This suggests that wolframin and SERCA function together in the same direction to regulate sleep, while RyR and IP3R act in the opposite direction.

## Discussion

Sleep problems have been reported in WS1 patients [8]. Their scores on Pediatric Sleep Questionnaire are more than 3 times higher than healthy controls and doubled compared to individuals with type I diabetes, indicating that the sleep issues are not merely due to metabolic disruptions. Indeed, our work here suggests that the sleep problems in human patients are of neural origin, specifically in the wake-promoting Dop2R neurons. Given that the rebound sleep is not significantly altered in *wfs1* depleted flies, we believe that lack of *wfs1* does not shorten sleep duration by impairing the sleep homeostasis system. Instead, *wfs1* deficiency leads to excessive wakefulness which in turn results in decreased sleep. Considering that heterozygous *WFS1* mutation is present in up to 1% of the population, it would be interesting to examine whether these heterozygous mutations contribute to sleep disruptions in the general population [9].

In mouse, chick, quail and turtle, *Wfs1* has been shown to be expressed in brain regions where dopamine receptor *Drd1* is expressed [51]. D1-like dopamine receptor binding is increased while striatal dopamine release is decreased in *Wfs1^{-/-}* mice [51–53]. Our results here also implicate a role for wolframin in dopamine receptor neurons and that lack of *wfs1* impacts dopaminergic signaling, as the effects of *wfs1* deficiency on both sleep and MB calcium concentration is blocked by the tyrosine hydroxylase inhibitor AMPT. Both *Dop2R*GAL4 and $G_o\alpha$GAL4 exhibit prominent expression in the MB, and to be more specific, in the α and β lobes of MB (S7C and S7D Fig). Previous studies have shown that dopaminergic neurons innervate wake promoting MB neurons, and here we found Dop2R and $G_o\alpha$+ cells to be wake-promoting as well [54]. Therefore, we suspect that wolframin functions in MB Dop2R/$G_o\alpha$+ neurons to influence sleep. Taken together, these findings suggest an evolutionarily conserved role of wolframin within the dopaminergic system. As this system is also important for sleep/wake regulation in mammals, it is reasonable to suspect that wolframin functions in mammals to modulate sleep by influencing the dopaminergic tone as well [55].

MB neural activity appears to be enhanced in *wfs1* deficient flies based on the results obtained using CaLexA and spH reporters. This elevated activity is consistent with our behavioral data, as activation of Dop2R/$G_o\alpha$+ cells reduces sleep, similar to the effects of *wfs1* deficiency. Moreover, silencing Dop2R neurons rescues the short-sleep phenotype of *wfs1* mutants, while over-expressing *wfs1* restores the decreased sleep induced by activation of Dop2R neurons. These findings suggest that wolframin functions to suppress the excitability of MB Dop2R neurons, which in turn reduces wakefulness and promotes sleep. Comparable cellular changes have been observed in *SERCA* mutant flies. Electric stimulation leads to an initial increase followed by prolonged decrease of calcium concentration in mutant motor nerve terminal compared to the control, while action potential firing is increased in the

mutants [56]. This series of results underpin the importance of ER calcium homeostasis in determining membrane excitability and thus neural function.

GCaMP6 monitoring reveals that *wfs1* deficiency selectively reduces fluorescence signal in the MB both under baseline condition and after dopamine treatment, which should reflect a reduction of cytosolic calcium level that is usually associated with decreased excitability [45]. Previous studies have shown that lack of wolframin leads to increased basal calcium level in neural progenitor cells derived from induced pluripotent stem (iPS) cells of WS1 patients and primary rat cortical neurons, but after stimulation the rise of calcium concentration is smaller in *Wfs1* deficient neurons, resulting in reduced calcium level compared to controls [12,46]. Similarly, evoked calcium increase is also diminished in fibroblasts of WS1 patients and MIN6 insulinoma cells with *WFS1* knocked down [13,50]. Notably, wolframin has been shown to bind to calmodulin (CaM) in rat brain, and is capable of binding with calcium/CaM complex *in vitro* and in transfected cells [57]. This may undermine the validity of using GCaMP to monitor calcium level in *wfs1* deficient animals and cells, and could potentially account for the contradictory data acquired using CaLexA vs GCaMP.

It is intriguing that in our study here *wfs1* deficiency appears to selectively impair the function of Dop2R/$G_o\alpha$+ neurons. It has been shown that in the rodent brain *Wfs1* is enriched in layer II/III of the cerebral cortex, CA1 field of the hippocampus, central extended amygdala, striatum, and various sensory and motor nuclei in the brainstem [58–61]. *Wfs1* expression starts to appear during late embryonic development in dorsal striatum and amygdala, and the expression quickly expands to other regions of the brain at birth [61]. We suspect that in flies *wfs1* may be enriched in Dop2R/$G_o\alpha$+ cells during a critical developmental period, and that sufficient level of wolframin is required for their maturation and normal functioning in adults. Another possibility is that these cells are particularly susceptible to calcium dyshomeostasis induced by loss of *wfs1*. Indeed, this is believed to be an important cause of selective dopaminergic neuron loss in Parkinson's Disease, as dopaminergic neurons are unique in their autonomic excitability and selective dependence on calcium channel rather than sodium channel for action potential generation [62]. We reason that Dop2R/$G_o\alpha$+ neurons may also be more sensitive to abnormal intracellular calcium concentration, making them particularly vulnerable to *wfs1* deficiency. The pathogenic mechanism underlying the neurodegeneration of WS1 is quite complex, possibly involving brain-wide neurodegenerative processes and neurodevelopmental dis-regulations [4]. Our findings here provide some evidence supporting a role for altered dopaminergic system during development. Obviously, much more needs to be done to test these hypotheses.

The precise role of wolframin in ER calcium handling is not yet clear. It has been shown in human embryonic kidney (HEK) 293 cells that knocking down *WFS1* reduces while overexpressing *WFS1* increases ER calcium level [63]. The authors concluded that wolframin upregulates ER calcium concentration by increasing the rate of calcium uptake. Consistently, we found by genetic interaction that knocking down *RyR or Itpr* (which act to reduce ER calcium level and thus knocking down either one will increase ER calcium level) rescues the short-sleep phenotype caused by *wfs1* mutation, while knocking down *SERCA* (which acts to increase ER calcium level and thus knocking down this gene will reduce ER calcium level) synergistically enhances the short-sleep phenotype. Based on the results of these genetic interactions, we propose that lack of *wfs1* increases cytosolic calcium while decreasing ER calcium, leading to hyperexcitability of Dop2R neurons and thus reduced sleep (Fig 7F). Knocking down *RyR* or *Itpr* decreases cytosolic calcium and increases ER calcium, counteracting the influences of *wfs1* deficiency and thus rescuing the short-sleep phenotype. On the other hand, knocking down *SERCA* further increases cytosolic calcium and decreases ER calcium, rendering an enhancement of the short-sleep phenotype. In line with this, study conducted in neural

progenitor cells derived from iPS cells of WS1 patients demonstrated that pharmacological inhibition of RyR can prevent cell death caused by *WFS1* mutation [46]. In addition, inhibiting the function of IP3R may mitigate ER stress in wolframin deficient cells [47,48]. One caveat is that SERCA protein level is increased in primary islets isolated from *Wfs1* conditional knock-out mice, as well as in MIN6 cells and neuroblastoma cell line with *WFS1* knocked down [50]. We reason this may be a compensatory increase to make up for the reduced ER calcium level due to wolframin deficiency. We do acknowledge that the hypothesis proposed in Fig 7F is not supported by our GCaMP data, which indicates decreased cytosolic calcium level in Dop2R neurons of *wfs1* deficient flies. We suspect that since the sleep phenotype associated with lack of *wfs1* is of developmental origin, it is possible there is an initial increase of cytosolic calcium during critical developmental period in *wfs1* deficient flies and this influences the function of Dop2R neurons in adults. Clearly, further characterizations need to be done to fully elucidate this issue, and preferably another calcium indicator independent of the GCaMP system should be employed.

In conclusion, here we identify a role for wolframin in the wake-promoting Dop2R neurons. *wfs1* depletion in these cells lead to impaired calcium homeostasis and altered neural activity, which in turn leads to enhanced wakefulness and reduced sleep. Our study may provide some insights for the mechanisms underlying the sleep disruptions in individuals with *WFS1* mutation, as well as for the pathogenesis of WS1.

## Materials and methods

### Fly strains

Flies were raised on standard cornmeal food at 25˚C and ~50% humidity under 12h light/12h dark (LD) cycles. All strains were obtained from Bloomington Stock Center TsingHua Fly Center, and Vienna *Drosophila* Resource Center, or as gifts from colleagues. All fly crosses were carried out at 25˚C unless noted otherwise. The following fly strains were used in this study: *Dop2R*GAL4, *Dop1R1*GAL4, *Dop1R2*GAL4, *DAT*GAL4 and *TH*GAL4 which were generated in Dr. Yi Rao's laboratory [30], UAS-*wfs1*RNAi#1 (TH02454.N), UAS-*wfs1*RNAi#2 (TH201500967.S), *wfs1*$^{MI14041}$ (BDSC:59250), UAS*wfs1#1* (BDSC: 8357), UAS*wfs1#2* (BDSC: 8356), UAS-*GFP*RNAi (BDSC: 9331), UAS-*ple*RNAi (BDSC: 25796), UAS-*Dop2R*RNAi (VDRC: V11471), UAS*NachBac* (BDSC:9467), UAS-*shi*$^{ts1}$(BDSC: 44222), UAS*TrpA1* (BDSC: 26263), UAS-*SERCA*RNAi#1 (THU2107), UAS-*SERCA*RNAi#2 (THU5676), UAS-*RyR*R-NAi#1 (TH02531.N), UAS-*RyR*RNAi#2 (THU5738), UAS-*Itpr*RNAi (BDSC: 25937), isogenic *w*$^{1118}$ (BDSC: 5905), UASmLexA-VP16-*NFAT* (BDSC: 66542), UAS*spH* [41], UASGCaMP6m (BDSC: 42750), UASmCD8GFP (BDSC: 32186), UASmCD8RFP (BDSC: 32219), *repo*GAL4 (BDSC: 7415), *elav*GAL4 (BDSC: 458), *tub*GAL4 [64], *tim*GAL4 (BDSC: 7126), *cry*GAL4-16 (BDSC: 24514), $G_o\alpha$GAL4 (BDSC: 104410), 247GAL80 (BDSC: 64306), UAS*dcr2* (BDSC: 24650), *tub*GAL80$^{ts}$#1 (BDSC: 7017) and *tub*GAL80$^{ts}$#2 (BDSC: 7019). All flies used for sleep monitoring were backcrossed with the isogenic *w*$^{1118}$ strain for at least 5 times. All experiments were conducted in male flies unless otherwise specified.

### Sleep and locomotor rhythm analysis

*Drosophila* Activity Monitor system (Trikinetics) was used to analyze fly sleep and locomotor rhythm. Flies 3~5 days old were used for experiments unless otherwise specified. Flies were entrained under LD at 25˚C for 3 days, and then their activities in the next 4 days under LD condition were analyzed, followed by 7 days of DD. Sleep is defined as 5 min consecutive inactivity. Sleep was analyzed with Counting Macro written in Excel (Microsoft) following previously published protocol [65]. Waking activity is calculated by dividing daily total activity

during waking period by the daily duration of waking period. Chi-squared periodogram analyses of period, power, and significance values during DD was carried out using ClockLab (Actimetrics) software. For DD rhythmicity, rhythmic flies were defined as those with chi-squared power-significance ≥10. %Rhythmic is calculated as the percentage of flies of a given genotype that exhibit power-significance ≥10. Period calculations considered all flies with power-significance ≥10.

To knock down *wfs1* only in adult stage, flies were raised at 19˚C and sleep and locomotor rhythm were monitored at 27–29˚C. To knock down *wfs1* only in developmental stage, flies were raised at 29˚C and sleep and locomotor rhythm were monitored at 19˚C. *TrpA1*, *NachBac* and *shi^{ts1}* flies were raised at 19˚C and baseline sleep was monitored at 19˚C. Temperature was then raised at lights on to 25.4˚C (for *TrpA1*), 27˚C (for *NachBac*) and 29˚C (for *shi^{ts}*) for further sleep monitoring.

## Sleep Deprivation

Mechanical sleep deprivation was performed at 25˚C using a multi-tube vortexer (VWR) modified by TriKinetics to house DAM2 activity monitors. After 3 days of LD entrainment and 1 day of baseline sleep recording, the multi-tube vortexer delivered 10s-long vibrations at random intervals centered around 60 s (± 30 s). The intensity of the vortexer was set to 4. Flies that displayed sleep loss ≥ 90% during the day of sleep deprivation were used to analyze the recovery sleep. The amount of sleep lost was calculated by subtracting the sleep duration during sleep deprivation from that of baseline sleep. The sleep loss percentage was calculated by amount of sleep lost dividing baseline sleep duration. Recovery sleep was calculated by the sleep duration in the first two hours of the day after deprivation subtracting that of the day before deprivation.

## Drug treatment

For pharmacological experiments, drugs were mixed in the fly food at indicated concentrations and the flies were fed with these food for the entire period during sleep monitoring. Concentrations of different drugs are as follows: 0.5–2 mg/ml AMPT (Sigma), 5 mg/ml L-DOPA (BBI Life Sciences), or 10 Mm ethanolamine-O-sulphate (Tokyo Chemical Industry), 10 mg/ml nipecotic acid (Sigma), 5 mM riluzole (Tokyo Chemical Industry), and 1% alcohol as a vehicle control for riluzole (Sinopharm Chemical Reagent Co., Ltd).

## ELISA

30 fly heads per sample were homogenized for 1 min with an automated tissue homogenizer (Shanghai Jinxin) at 4˚C in 300 μl lysis buffer (0.01N HCl, 1mM EDTA, 4mM sodium metabisulfite). The samples were then centrifuged at 13,680 *g* for 30 min at 4˚C and supernatants were collected. 100 μl supernatant was used per reaction to measure dopamine level using Dopamine Research ELISA following the manufacturer's instructions (Labor Diagnostika Nord).

## RNA Extraction and Quantitative Real-Time PCR (qRT-PCR)

Approximately 20–30 7-day-old flies were collected and frozen immediately on dry ice. For analysis of clock gene expression, flies were entrained for at least 3 days in LD and then collected at the indicated time points. Fly heads were isolated and homogenized in Trizol reagent (Life Technologies). Total RNA was extracted and qRT-PCR conducted following our previously published procedures [66]. The sequences of primers used are in S4 Table.

## Confocal imaging

For live imaging experiments (GCaMP and spH), flies 2–3 days old were collected into tubes with standard food and maintained under LD for 3 days. For AMPT treatment, flies 2–3 days old were treated with 2mM AMPT for 3 days in LD. On day 4, flies were anesthetized with $CO_2$ and brains were dissected at the indicated time points in PBS. The dissected brain samples were transferred into *Drosophila* adult hemolymph-like saline solution on glass slide and sealed with cover slide. The duration for dissection and microscopy should be completed within 2 hours for each time point. Images were captured with Olympus FV3000 confocal microscopy with a 20X objective lens. Time-series images were collected within a period of 250 seconds at 4 Hz, and the resolution was set as 256 X 256 pixels. After capturing baseline images, dopamine hydrochloride (Sigma, 2 mM) was applied using a pipette. Maximal GCaMP intensity was normalized to that of the baseline.

For immunostaining and GFP imaging, flies were anesthetized with $CO_2$ and dissected and fixed with 4% paraformaldehyde diluted in PBS. The brains were then fixed with 4% paraformaldehyde for 30 minutes. Samples were washed with 1×PBT (1×PBS + 0.3% TritonX-100) for 3 times and dissected further to remove additional debris in 1×PBS solution. For immunostaining, brain samples were then blocked in 1×PBT solution with 5% fetal bovine serum (Hyclone) for 30 minutes and subsequently stained with mouse anti-nc82 (1:100, DSHB). After 3×10 min PBT rinses, the brains were incubated with goat anti-mouse-Cy5 (1:500, Abcam) at room temperature for 2 h. Then the brains were rinsed 3×10 min in PBS and mounted and imaged using Olympus FV3000 confocal microscope with a 20X objective lens.

Images were acquired using the same settings (power, gain, offset) for each experiment. GCaMP, GFP, spH and RFP fluorescence intensity were measured and quantified by ImageJ software. The mushroom body and antennal lobe border was traced manually.

## Statistical analysis

One-way ANOVA (Prism Graphpad) was used to compare the differences between multiple genotypes. For data that fit normal distribution, Student's t-test (Prism Graphpad) was used to compare the difference between two genotypes. For data that do not fit normal distribution, Mann-Whitney test (Prism Graphpad) was used to compare the difference between two genotypes.

## Supporting information

**S1 Fig. Validation of *wfs1* knock-down and over-expression.** (A) and (C) Relative mRNA abundance of *wfs1* determined by qRT–PCR in whole-head extracts. (B) Daily sleep duration of GFP RNAi flies. n = 20–35. For comparison between RNAi flies vs. UAS/GAL4 controls, one-way ANOVA was used: compared to GAL4 control, #$P < 0.05$; compared to UAS control, ***$P < 0.001$. For (A) and (C), the value of GAL4 control at was set to 1. n = 4–6. Error bars represent SEM. Mann-Whitney test: compared to GAL4 control, #$P < 0.05$, ##$P < 0.01$; compared to UAS control, *$P < 0.05$, **$P < 0.01$. G4, GAL4; U, UAS.
(TIF)

**S2 Fig. Knocking down *wfs1* during development reduces adult sleep duration.** Daily sleep duration of flies with *wfs1* knocked down in adult stage (A) or developmental (B) stage. n = 31–126. For comparison between RNAi flies vs. UAS/GAL4 controls, one-way ANOVA was used: compared to GAL4 control, #$P < 0.05$, ###$P < 0.001$; compared to UAS control, **$P < 0.01$, ***$P < 0.001$. G4, GAL4; U, UAS; G80, GAL80.
(TIF)

**S3 Fig.** *Wfs1* **deficiency leads to reduced sleep duration in aged flies.** (A, C and E) The sleep profile of *wfs1* RNAi flies, *wfs1* mutants and controls collected 30 days post-eclosion. (B, D and F) The sleep profile of flies, *wfs1* mutants and controls collected 30 days post-eclosion. Gray shade indicates the dark period. For comparison between RNAi flies vs. UAS/GAL4 controls, one-way ANOVA was used: compared to GAL4 control, #$P < 0.05$, ###$P < 0.001$; compared to UAS control, **$P < 0.01$, ***$P < 0.001$. For comparison between mutant vs. control, Mann-Whitney test or Student's t-test was used: ***$P < 0.001$. n = 26–73. Error bars represent SEM. G4, GAL4; U, UAS; *M*, *wfs1*[MI14041]; ZT, Zeitgeber Time.
(TIF)

**S4 Fig. Pan-neuronal depletion of** *wfs1* **does not significantly alter recovery sleep.** (A) Sleep profile of *wfs1* RNAi and control flies the day before, during and after sleep mechanical deprivation. White and black bars indicate light and dark period, respectively. (B) Left panel: sleep duration the day before sleep deprivation. Middle panel: percentage of sleep lost on the day of sleep deprivation. Right panel: The recovery sleep during the first 2 h on the day immediately after sleep deprivation. One-way ANOVA: compared to GAL4 control, ###$P < 0.001$; compared to UAS control, *$P < 0.05$, **$P < 0.01$, ***$P < 0.001$. n = 18–49. Error bars represent SEM. G4, GAL4; U, UAS; ZT, Zeitgeber Time.
(TIF)

**S5 Fig. The effects of pharmacological treatments on the sleep duration of** *wfs1* **deficient flies.** Daily sleep duration of *wfs1* RNAi, mutant and control flies treated with the indicated drugs. Alcohol is used as a vehicle control for riluzole, while all other drugs were directly dissolved in the food. For comparison between RNAi flies vs. UAS/GAL4 controls, one-way ANOVA was used. For comparison between mutants expressing RNAi vs. mutant control, Student's t-test was used. ***$P < 0.001$. n = 21–67. Error bars represent SEM. G4, GAL4; U, UAS; *M*, *wfs1*[MI14041]; NipA, nipecotic acid; EOS, ethanolamine-O-sulphate.
(TIF)

**S6 Fig. Pan-neuronal depletion of** *wfs1* **does not significantly alter the expression of genes involved in dopamine signaling or dopamine concentration.** (A-G) Relative mRNA abundance of genes involved in dopamine signaling in whole-head extracts, determined by qRT-PCR. For each experiment, the value of G4 control was set to 1. (H) Dopamine concentration in whole head extracts determined by ELISA. Error bars represent SEM. n = 3–6. Mann-Whitney test: compared to GAL4 control, #$P < 0.05$. G4, GAL4; U, UAS; *M*, *wfs1*[MI14041].
(TIF)

**S7 Fig. Expression pattern of Dop2RGAL4 and G$_o$αGAL4 in the brain. (A and B)** Representative image of adult fly brain with mCD8GFP expression in Dop2R (A) or G$_o$α+ (B) cells. (C) and (D) are enlarged image of MB shown in (A) and (C), respectively. The brains are stained with antibody against BRUCHPILOT (NC82) to label axons. Red arrowhead indicates the MB. The scale bar represents 50 μm and 17 μm, respectively. *D2R*G4, *Dop2R*GAL4; G4, GAL4; U, UAS.
(TIF)

**S8 Fig.** *Wfs1* **depletion reduces the GCaMP signal of MB Dop2R neurons.** Representative live image of adult fly brain with pan-neuronal expression of GCaMP6m and mCD8RFP. Brain samples were collected and dissected at the indicated time points. (B) Quantification of GCaMP6m/RFP signal intensity in MB (two-way ANOVA, n = 16–20). (C) Left panel: representative live image of adult fly brain with pan-neuronal expression of GCaMP6m and

mCD8RFP with the antennal lobe indicated by the white rectangle. Brain samples were collected and dissected at ZT6. Right panel: quantification of antennal lobe GCaMP6m/RFP intensity (one-way ANOVA, n = 22, 20, 22). (D) Left panel: representative live image of adult fly brain with pan-neuronal expression of GCaMP6m and mCD8RFP. The flies were fed with 2 mM AMPT for 3 days. Brain samples were collected and dissected at ZT6. Right panel: quantification of GCaMP6m/RFP signal intensity in MB (two-way ANOVA, n = 20). (E) Left panel: representative live image of adult fly brain with GCaMP6m expressed in Dop2R neurons. Brain samples were collected and dissected at ZT6. Right panel: quantification of MB GCaMP6m signal intensity normalized to the control (Student's t-test, n = 26, 28). (F) Left panel: time-series GCaMP6m intensity of adult fly brain treated with dopamine. GCaMP6m intensity was normalized to the baseline level. Right panel: quantification of maximum GCaMP6m intensity after dopamine treatment (Student's t-test, n = 17, 10). Error bars represent SEM. $^*P < 0.05$, $^{**}P < 0.01$, $^{***}P < 0.001$. The scale bar represents 50 μm unless indicated otherwise. G4, GAL4; U, UAS; *D2R*GAL4, *Dop2R*GAL4. ZT, Zeitgeber Time.
(TIF)

**S9 Fig. *Wfs1* deficiency in the MB contribute to the reduced sleep duration.** (A) Representative image of adult fly brain with mCD8GFP expression in non-MB Dop2R neurons. The brains are stained with antibody against BRUCHPILOT (NC82) to label axons. The scale bar represents 50 μm. (B) Daily sleep duration of *wfs1*RNAi flies and controls. For comparison between RNAi flies vs. UAS/GAL4 controls, one-way ANOVA was used: compared to UAS control, $^{**}P < 0.01$, $^{***}P < 0.001$. n = 18–39. Error bars represent SEM. *D2R*G4, *Dop2R*GAL4; U, UAS; G80, GAL80.
(TIF)

**S10 Fig. *Wfs1* heterozygous mutation does not influence the sleep of flies with active Dop2R neurons.** (A and B) Daily sleep duration (A) and wake activity (B) of *wfs1* heterozygous mutant flies with Dop2R neurons activated by TrpA1. For comparison between RNAi flies vs. UAS/GAL4 controls, one-way ANOVA was used: $^{**}P < 0.01$, $^{***}P < 0.001$. For comparison between mutants expressing TrpA1 vs. mutant control, Student's t-test was used: $^{***}P < 0.001$. n = 24–45. Error bars represent SEM. *D2R*G4, *Dop2R*GAL4; U, UAS; M, *wfs1^{MI1404}*.
(TIF)

**S11 Fig. Validation of *RyR* and *SERCA* knock-down.** (A and B) Relative mRNA abundance of *RyR* (A) and *SERCA* (B) determined by qRT-PCR in whole-head extracts. Female fly heads were used for this entire experiment as we were not able to obtain male *elav*GAL4;U*RyR*RNAi#1/+ flies. We were not able to obtain either male or female *elav*GAL4;U*SERCA*RNAi#2/+ flies. For each experiment, the value of G4 control at was set to 1. Error bars represent SEM. n = 3–5. Mann-Whitney test: $^*P < 0.05$, $^{**}P < 0.01$. G4, GAL4; U, UAS.
(TIF)

**S12 Fig. Knocking down *Itpr* lengthens the sleep duration of *wfs1* mutant flies.** Daily sleep duration of *wfs1* mutant flies with *Itpr* knocked down in all neurons. For comparison between RNAi flies vs. UAS/GAL4 controls, one-way ANOVA was used. For comparison between mutants expressing RNAi vs. mutant control, Student's t-test was used: $^{**}P < 0.01$, $^{***}P < 0.001$. n = 38–116. Error bars represent SEM. *e*G4, *elav*GAL4; U, UAS; M, *wfs1^{MI14041}*.
(TIF)

**S1 Movie. Time series GCaMP6m of WT fly brain treated with dopamine, related to S8F Fig.**
(AVI)

**S2 Movie. Time series GCaMP6m of *wfs1RNAi* fly brain treated with dopamine, related to S8F Fig.**
(AVI)

**S1 Data. Data for Reconstruction of Figures.**
(XLSX)

**S1 Table. Locomotor rhythm of flies with *wfs1* knocked down during adult or development.**
(DOCX)

**S2 Table. *Wfs1* deficiency dampens locomotor rhythm in aged flies.**
(DOCX)

**S3 Table. The locomotor rhythm of *wfs1* mutant flies with *ple* or *Dop2R* knocked down.**
(DOCX)

**S4 Table. Oligonucleotides used in this study.**
(DOCX)

## Acknowledgments

We would like to thank Drs. Magaret Ho, Yi Rao, Liming Wang, Junhai Han and Yi Zhong for kindly providing flies used in this study.

## Author Contributions

**Conceptualization:** Luoying Zhang.

**Investigation:** Huanfeng Hao, Li Song.

**Supervision:** Luoying Zhang.

**Writing – original draft:** Huanfeng Hao, Li Song, Luoying Zhang.

**Writing – review & editing:** Huanfeng Hao, Li Song, Luoying Zhang.

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
