## [Decision Letter · Decision Letter 0]

29 Nov 2022

Dear Dr Zhang,

Thank you very much for submitting your Research Article entitled 'Wolfram syndrome 1 regulates sleep in dopamine receptor neurons by modulating calcium homeostasis' to PLOS Genetics.

The manuscript was fully evaluated at the editorial level and by independent peer reviewers. The reviewers appreciated the attention to an important problem, but raised some substantial concerns about the current manuscript. In particular they ask for more clarity regarding where the gene is acting to exert its effect on sleep, and the role of calcium in producing the defects caused by mutations in wfs1. Based on the reviews, we will not be able to accept this version of the manuscript, but we would be willing to review a much-revised version. We cannot, of course, promise publication at that time.

If you decide to revise the manuscript for further consideration at PLOS Genetics, please aim to resubmit within the next 60 days, unless it will take extra time to address the concerns of the reviewers, in which case we would appreciate an expected resubmission date by email to plosgenetics@plos.org.

We are sorry that we cannot be more positive about your manuscript at this stage. Please do not hesitate to contact us if you have any concerns or questions.

Yours sincerely,

John Ewer

Academic Editor

PLOS Genetics

Hua Tang

Section Editor

PLOS Genetics

Reviewer's Responses to Questions

**Comments to the Authors:**

Reviewer #1: The authors found that in Drosophila, the inhibition of wfs1 causes increased arousal, resulting in reduced sleep. This phenotype was shown to involve dopamine signaling along with ER Ca2+ dynamics. Altered ER Ca2+ dynamics in wfs1 mutations has already been reported, but this manuscript may be the first report to show a sleep behavior in flies. Each experiment was carefully analyzed. However, there are some problems with the in vivo experiments, which are listed below.

The biggest problem is the Ca2+ imaging experiment in Fig. 5. These are all observed at one time point after dissection to estimate basal Ca2+ concentration and Ca2+ concentration during neural activity. These methods are indirect and it is questionable whether they accurately measure neural activity-dependent Ca2+ dynamics. Ca2+ dynamics before and after dopamine (DA) stimulation should be shown in the same samples.

The authors seem to focus on Ca2+ alteration in the mushroom body (MB). However, it is possible that ER Ca2+ alteration occur in both DA and MB neurons, considering that AMPT ameliorated the phenotypes. The authors should confirm the possibility of DA neurons overactivity, DA hypersecretion, and tyrosine hydroxylase upregulation and Ca2+ alterations should also be confirmed in wfs1 mutant DA neurons.

The authors showed that knockdown of RyR rescues the phenotype, but why not also consider the involvement of IP3R?

In Table 1, isn't it an important phenotype that the power of locomotor rhythm increases with Repo drive?

The authors should also analyze the effects of aging on Ca2+ dynamics, behavioral abnormalities, and neurodegeneration.

Minor comments.

Please indicate directly whether Dop2R+ and Goα+ cells analyzed here are the same neurons.

The labeling in Fig 3C are difficult to understand, please devise new labels. It is also difficult to understand which one is significantly different from the other.

The definitions of wake activity, the power of locomotor rhythm and % rhythm in figures and table are not clear.

The graphs in Fig 4 is difficult to understand due to lack of explanation.

The sentence "On the other hand, waking activity is not increased when these neurons are activated, which means the enhanced wakefulness is not merely due to hyper activity (Fig 4C)." should rather describe "rather decreased."

Reviewer #2: This work by Hao et al examines the function of wolframin in the the regulation of sleep-relevant circuits, providing insight into possible mechanisms underlying sleep disorders. Given that dopamine is a conserved regulator of arousal, the findings could be generally relevant to mammals. The authors use multiple, independent genetic approaches to describe the wolframin-deficient sleep phenotype, and use a combination of drug experiments cell-type-specific rescue/KD experiments to show that wfs acts in dopaminergic neurons to regulate their activity. They go on to show that wfs influences sleep by regulating Ca homeostasis, thus providing mechanistic insight into the phenotype. While the manuscript covers a lot of ground, there are a number of areas that are confusing and require further experiments to clarify.

While the authors carry out rigorous genetic experiments to support most of their conclusions, a major conclusion of the work is that wfs negatively regulates the activity of Dop2R neurons to regulate sleep duration. These conclusions are based on Ca imaging studies whose interpretations are made difficult by two points:

-Loss of wfs1 causes opposite phenotypes using GCamP and Calexa.

-Relevance of MB lobes to sleep phenotypes not established

Given conflicting results using Ca imaging, I would have stronger faith in conclusions if authors provided additional genetic experiments to test (1) hypothesis that wfs1 limits activity of Dop2R neurons and (2) relevance of MB projecting dop2R cells to phenotype. Major revisions should include addressing:

1. Despite their pharmacologic experiments, their "cell-type-specific" GAL4 experiments for Dop2R neurons are difficult to interpret. The Dop2R-GAL4 is quite broadly expressed as is Galphao. A major conclusion of the paper is that the MB is relevant for the sleep phenotype. The authors should show this directly using intersectional approaches to label subsets of Dop2R cells projecting to MB in RNAi and/or rescue experiments. Other experiments to consider would be depletion or reduction of dop2R receptor alongside reduction of wfs1 activity, and/or silencing (e.g. Kir, shi-DN, tetanus toxin) of dop2R cells with reduced wfs1.

2. The data suggest that wfs1 is acting in an ongoing manner to regulate sleep. The authors should show that RNAi expression only the adult leads to the phenotype.

3. Figure 5 is very confusing and I do not fully agree with the interpretation of gcamp v calexa results. Calexa is a Ca2+ based reporter and I would expect that the results should match gcamp. For this figure to hold up, the authors need to do more to tease apart the differential results. The mixed findings in figure 6 likewise are confusing and fail to clarify - based on these results, the authors draw conclusions about genetic interactions but fail to explain a model that takes the differential results into account. This area of the manuscript requires the most work, as in current form Figures 5-6 are quite inconclusive.

More specific minor points on each figure follow:

Figure 1

1. All graphs in this figure should show individual data points to provide a more clear sense of data spread

Fig 3

1. Presentation of data in A is quite confusing: is AMPT treatment normalizing differences between genotypes, rescuing sleep amount, or both? Preferably should be both. Sleep values on y axis look smaller than those shown for non-drug experiments in Fig 1 and in 3B. For clarity, should include data showing effect +/- drug on genetic controls (predict sleep increase).

2. Data as shown in 3A, the rescue looks complete. Is this rescue dose-dependent? Can feeding flies L-DOPA exacerbate phenotype? Can they rescue using an independent genetic approach: KD TH using Th-GAL4 in mutant. Not satisfied with just one drug experiment.

3. 3C: very hard to follow this figure, is there a way to make it easier to read?

Fig 4

1. Even though the dop2R>TrpA1 experiment has not been published to my knowledge, the dop2R null phenotype is known: sleepy flies. So the idea that Dop2R cells are wake-promoting is not new.

For what it's worth, I think the very strong wake-promoting effects seen even with gentle TrpA1 activation probably reflect the broad expression pattern of the GAL4s used.

2. There are very pronounced effects of locomotor activity in dop2R>TrpA1 and Gao>TrpA1. While I appreciate that these locomotor phenotypes are separable from sleep phenotypes (4B, 19C dark period), I would appreciate more clear description of activity phenotypes. The effects are probably light-dependent because Dop2R activity is known to be light gated.

3. 4D-F: not sure how to match up bar graphs with sleep traces. Should make this more clear.

4. Since wfs in Dop2R+ cells promotes sleep, what would happen to dop2R>TrpA1 phenotype in : a. M/+ b. wfs1 overepxression?

Fig 5--Obviously very confusing w/ different Ca reporters, but also unclear about relevance of MB to wfs LOF sleep phenotypes.

1. Are these brains kept in ex vivo preps for 18h? Or are they dissecting separate brains at separate time points? Based on legend, think it's the latter. In that case, not appropriate to display data in 5B as though these were repeated measures of the same brains.

2. authors switch arbitrarily between reporting GFP/RFP ratios to just GFP intensity. Should be consistent across all experiments, especially for Dop2R-GAL4.

3. 5F,G: This is confusing. Phenotype looks so dramatic. Yet how do we interpret the reduced sleep in dop2R/elavG4>wfs-IR flies?

Reviewer #3: In this study, authors investigated a role of WOLFRAM SYNDROME 1 (WFS1) deficiency in sleep regulation using a Drosophila model since heterozygous mutation carriers of WFS1 exhibit higher risk of psychological disorders including sleep disturbance via unknown mechanisms. They found that knocking down wfs1, a fly ortholog of WFS1, in neurons or wfs1 mutation reduced sleep. Through genetic and pharmacologic approaches, they also demonstrated that a lack of wfs1 in dopamine 2-like receptor (Dop2R) expressing neurons was responsible for observed effects through dopaminergic signaling. As for a mechanism, they proposed that wfs1 deficiency decreased cytosolic calcium via disruption of ER-mediated calcium homeostasis and increased the excitability of Dop2R expressing neurons.

This is an interesting study to add novel function of wfs1 in sleep regulation. This reviewer has several comments on the experimental design and interpretation of their results.

1. For experiments using wfs1 RNAi, it is better to used control RNAi as a control for RNAi expression.

2. Authors knocked down wfs1 in several different neurons including Dop2R expressing neurons and circadian clock neurons. Interestingly, however, wfs1 knockdown affected functions of Dop2R expressing neurons but not those of clock expressing neurons. Since wfs1 deficiency seemed to affect ER calcium homeostasis and neuronal excitability, it is not clear why Dop2R expressing neurons were specifically affected by wfs1 deficiency. Please clarify this point.

3. Authors used the calcium sensor GCaMP6m to monitor calcium levels in neurons and found that basal calcium levels were decreased by wfs1 knockdown. This result is interesting; however, a relationship between low calcium levels and high neuronal excitability is a little difficult to understand. It would be important to measure changes in calcium levels upon stimulation.

4. It would be informative to discuss the relevance of the findings to neuropathology in WS1 patients including dopaminergic system.

**Have all data underlying the figures and results presented in the manuscript been provided?**

Reviewer #1: **No: **The origin of all the fly strains used is not stated.

Reviewer #2: Yes

Reviewer #3: Yes

PLOS authors have the option to publish the peer review history of their article (what does this mean?). If published, this will include your full peer review and any attached files.

Reviewer #1: **Yes: **Yuzuru Imai

Reviewer #2: No

Reviewer #3: No

---

## [Decision Letter · Decision Letter 1]

1 Jun 2023

Dear Dr Zhang,

Thank you very much for submitting your revised Research Article entitled 'Wolfram syndrome 1 regulates sleep in dopamine receptor neurons by modulating calcium homeostasis' to PLOS Genetics.

The manuscript was fully evaluated at the editorial level and by independent peer reviewers. The reviewers appreciated the attention to an important topic but identified some concerns that we ask you address in a revised manuscript. In particular, both reviewers have serious reservations with the interpretation of the results using the calcium sensor. Please pay especial attention to this and, if necessary, please scale back the conclusions from these data.

We therefore ask you to modify the manuscript according to the review recommendations. Your revisions should address the specific points made by each reviewer.

Yours sincerely,

John Ewer

Academic Editor

PLOS Genetics

Hua Tang

Section Editor

PLOS Genetics

Reviewer's Responses to Questions

**Comments to the Authors:**

Reviewer #1: This reviewer is approximately satisfied with the authors' response. However, the interpretation of the GCaMP results remains a concern and the results do not support the hypothesis in Fig 7E.

The authors point out the possibility that wfs1 binds to the CaM region of GCaMP and affects GCaMP activity. However, this would need to be demonstrated experimentally. For example, it would be possible to overexpress wfs1 in unrelated tissues and see its effect on GCaMP activity. Or Ca2+ activity of neurons should be measured in modalities other than GCaMP.

Reviewer #2: The authors have done a tremendous amount of work in this revision, and attempted to address concerns of all the reviewers. While there are still some remaining issues with regard to interpretation of the calcium measures, I think this work is a nice addition to the literature. The new genetic expts (dop2R KD, dop2R neuron silencing) improve the rigor of the conclusions from the behavioral data. The GAL80ts result likewise adds a nice conclusion, showing that the sleep phenotype is of developmental origin. There are two issues that remain and should be addressed with changes to text.

1. The MB GAL80 experiments are not very convincing (Fig S9). At a minimum, the authors should say that MB GAL80 attenuates the phenotype (does not abolish it) and leave open the possibility that non MB regions contribute. One could argue that the authors should add the proper positive controls (full sleep phenotype) to this experiment to directly compare, as it is possible that in these particular experimental runs that the MB GAL80 does not impact the phenotype at all, meaning it is of non MB origin. Since this is unlikely, I would be satisfied with just a softening of the language - there is indeed still a sleep loss phenotype with RNAi in the setting of MB GAL80, perhaps just not as big.

2. I am still not satisfied with their interpretation of the Ca imaging data. The conclusions are not backed up by the experiments. I think the most telling result, which is somewhat buried in a supplemental figure, is that wfs is required for Dop2R neuronal activity in response to exogenous DA. This stands in direct contrast to idea of hyperexcitability being the underlying mechanism. But the authors call into question whether the results are reliable because of a potential interaction with wfs and GCaMP. While this is theoretically possible, I do not think it is sufficient grounds to disregard the data in favor of results that better fit their hypothesis. At this stage, however, I do not think it is reasonable to request more experiments. Instead, I favor really drawing back the conclusions and discussion as it pertains to hyperexcitability and the calcium results in general. The authors did a ton of work and this should be shown! But I think they can keep the discussion on issues with gcamp v calexa and have the idea of hyperexcitability as more speculative in the discussion since the results do not fully back this up.

**Have all data underlying the figures and results presented in the manuscript been provided?**

Reviewer #1: Yes

Reviewer #2: Yes

PLOS authors have the option to publish the peer review history of their article (what does this mean?). If published, this will include your full peer review and any attached files.

Reviewer #1: No

Reviewer #2: No

---

## [Decision Letter · Decision Letter 2]

13 Jun 2023

Dear Dr Zhang,

We are pleased to inform you that your manuscript entitled "Wolfram syndrome 1 regulates sleep in dopamine receptor neurons by modulating calcium homeostasis" has been editorially accepted for publication in PLOS Genetics. Congratulations!

Yours sincerely,

John Ewer

Academic Editor

PLOS Genetics

Hua Tang

Section Editor

PLOS Genetics

Comments from the reviewers (if applicable):

Reviewer's Responses to Questions

**Comments to the Authors:**

Reviewer #1: Regarding the GCaMP results, the authors have responded appropriately.

Reviewer #2: I am satisfied with the caveats added to the discussion and believe the work should be published without further experiments.

**Have all data underlying the figures and results presented in the manuscript been provided?**

Reviewer #1: Yes

Reviewer #2: Yes

PLOS authors have the option to publish the peer review history of their article (what does this mean?). If published, this will include your full peer review and any attached files.

Reviewer #1: No

Reviewer #2: No

**Data Deposition**

http://datadryad.org/submit?journalID=pgenetics&manu=PGENETICS-D-22-01232R2

**Press Queries**

---

## [Editor Report · Acceptance letter]

27 Jun 2023

PGENETICS-D-22-01232R2 

Wolfram syndrome 1 regulates sleep in dopamine receptor neurons by modulating calcium homeostasis 

Dear Dr Zhang, 

We are pleased to inform you that your manuscript entitled "Wolfram syndrome 1 regulates sleep in dopamine receptor neurons by modulating calcium homeostasis" has been formally accepted for publication in PLOS Genetics! Your manuscript is now with our production department and you will be notified of the publication date in due course.

With kind regards,

Zsuzsanna Gémesi

PLOS Genetics

On behalf of:
